# Warwick-India-Canada (WIC) global mental health group: rationale, design and protocol

Swaran P Singh [ID],[1,2] Mohapradeep Mohan [ID],[1] Srividya N Iyer,[3,4] Caroline Meyer,[5] Graeme Currie,[6] Jai Shah,[3,4] Jason Madan [ID],[7] Max Birchwood,[1] Mamta Sood,[8] Padmavati Ramachandran [ID],[9] Rakesh K Chadda,[8] Richard J Lilford,[10] Thara Rangaswamy,[9] Vivek Furtado,[1] on behalf of the WIC Consoritum

SPS and MM contributed equally.

For numbered affiliations see end of article.

**Correspondence to**
Professor Swaran P Singh;
s.p.singh@warwick.ac.uk

## ABSTRACT

**Introduction** The primary aim of the National Institute of Health Research-funded global health research group, Warwick-India-Canada (WIC), is to reduce the burden of psychotic disorders in India. India has a large pool of undetected and untreated patients with psychosis and a treatment gap exceeding 75%. Evidence-based packages of care have been piloted, but delivery of treatments still remains a challenge. Even when patients access treatment, there is minimal to no continuity of care. The overarching ambition of WIC programme is to improve patient outcomes through (1) developing culturally tailored clinical interventions, (2) early identification and timely treatment of individuals with mental illness and (3) improving access to care by exploiting the potential of digital technologies.

**Methods and analysis** This multicentre, multicomponent research programme, comprising five work packages and two cross-cutting themes, is being conducted at two sites in India: Schizophrenia Research Foundation, Chennai (South India) and All India Institute of Medical Sciences, New Delhi (North India). WIC will (1) develop and evaluate evidence-informed interventions for early and first-episode psychosis; (2) determine pathways of care for early psychosis; (3) investigate the efficacy and cost-effectiveness of community care models, including digital and mobile technologies; (4) develop strategies to reduce the burden of mental illnesses among youth; (5) assess the economic burden of psychosis on patients and their carers; and (6) determine the feasibility of an early intervention in psychosis programme in India.

**Ethics and dissemination** This study was approved by the University of Warwick's Biomedical and Scientific Research Ethics Committee (reference: REGO-2018-2208), Coventry, UK and research ethics committees of all participating organisations. Research findings will be disseminated through peer-reviewed scientific publications, presentations at learnt societies and visual media.

## Strengths and limitations of this study

► Warwick-India-Canada (WIC) will develop a bespoke clinical interventions with proven efficacy that is culturally adapted, appropriate, accessible and acceptable.

► This is the first multisite study to identify barriers, facilitators and economic evaluation for implementing early intervention services in psychosis in India.

► WIC will build research capacity with potential for future efforts, including in methodologies such as health economic evaluation and implementation science.

► WIC will identify pathways to mental healthcare in India and develop community-based models for continuity of clinical care.

► Some studies embedded are exploratory and hypothesis-generating; therefore, large studies may be warranted for conclusive evidence.

individuals develop their first episode of psychosis (FEP) during adolescence or early adulthood.[2] Importantly, the burden of disease and disability is particularly high in low/middle-income countries (LMICs).[3] LMICs have predominantly young populations, with the bulk of FEP cases receiving inadequate care.[3–5] Despite efforts to develop healthcare models for accessible and effective care in LMICs, the implementation of such models remains a major challenge.[6]

In the developed world, among the most significant reforms since deinstitutionalisation has been the establishment of early intervention in psychosis services (EIS).[7] Given their clinical effectiveness and cost-effectiveness,[8–10] many countries have scaled up EIS as part of their mental health policy. EIS focus on reducing the duration of untreated psychosis (DUP), provision of holistic and evidence-based specialist care

## INTRODUCTION

Globally, psychotic disorders such as schizophrenia are among the top 20 leading contributors to disability.[1] Nearly 80% of

during the 'critical period' of 2–5 years around onset of psychosis and interventions for individuals who are at high risk of developing a psychotic disorder (clinical high risk or ultra-high risk).[11 12] Despite such promising evidence, many LMICs have not implemented EIS into their mental health systems. Inadequate mental health workforce, fragmented healthcare systems[13] and scarcity of research and implementation capacity are significant barriers to introducing such care in LMICs.[14 15] Furthermore, DUP is significantly higher in LMICs,[16] and therefore EIS provision adapted to LMIC settings is potentially of great value. Help-seeking in LMICs is often pluralistic, with traditional practitioners being the initial and sometimes the only source of care.[17] Therefore, a simple 'transplantation' of the 'Western' EIS model to LMIC context is neither feasible nor practical. However, it may be possible to incorporate the principles and therapeutic components of EIS into locally contextualised mental healthcare in LMICs,.[3 18] This is the key premise underpinning the WIC project.

### Aim and strategic goals

The National Institute of Health Research (NIHR) Global Health Research Group (GHRG) on Psychosis Outcomes: the Warwick-India-Canada (WIC) network aims to reduce the burden of psychotic disorders and improve health, well-being and functioning of those with psychotic disorders in India. The WIC network brings together knowledge and expertise of four internationally recognised institutions that share research interests, expertise in effective interventions in psychotic disorders, strategic vision to reduce the burden of psychotic disorders in resource-poor settings and who are keen to expand their research knowledge to implement evidence-based clinical changes in LMIC settings. Informed by principles of evidence-based EIS developed in the UK and Canada, the WIC GHRG will develop culturally sensitive interventions that will expand mental health services to communities that currently have limited access in India. WIC will also build and sustain much-needed research capacity and support the implementation of evidence-based practice.

### The Indian context

India with its 1.3 billion population has a severe shortage of mental healthcare provision. Compared with a global average of 3.96 psychiatrists/100 000 population, India has 0.75 psychiatrists/100 000 population. The overall availability of mental healthcare workers (psychiatrists, clinical psycologists and psychiatric social workers) is <1/100 000 population.[19 20] India therefore has a large number of patients with undetected, untreated or inadequately treated psychotic disorders.[21] Even when patients access care, there is no continuity; treatment is sporadic and focused on symptomatic control and crisis management rather than recovery.[22] Social stigma and discrimination associated with mental illness are also major impediments to delivering evidence-based care.[23] Studies have found that many prodromal patients ('at-risk'

for psychosis) remain undetected in the community,[24] and there are few early identification programmes in operation.

### Objectives

The key objectives of WIC are the following:

1. To develop a culturally tailored FEP management protocol, based on evidence synthesis of current clinical practice at partner sites and by modifying protocol from the National Evaluation of Development of Early intervention Network (NEDEN) study, to make sustained health improvements for poorly served individuals with FEP.
2. To better understand pathways of care to develop improved early detection strategies for psychosis relevant to India.
3. To create a data set of well-characterised inception cohorts of FEP with research-ready systems and processes that will enable to answer new research questions.
4. To investigate clinical effectiveness and cost-effectiveness of innovative community care models for psychosis, including digital and mobile applications.
5. To develop and implement school-based and college-based intervention programmes for early detection and referral of individuals with psychosis.
6. To build research capacity by providing a strong multidisciplinary research and training programme for researchers based in India.

A larger goal is to build strong research collaborations between India, UK and Canada partners for long-term research, training, knowledge exchange and service improvement.

## METHODS

### WIC programme—design

The design of WIC programme is depicted in figure 1. Our strategic plan comprises five work packages (WPs) and two cross-cutting themes (CTs) with clear objectives and deliverables. In addition, we will exploit the potential of digital/mobile technologies to (1) identify mental disorders in students and (2) increase access to clinical services and continuity of care for difficult-to-treat cases of schizophrenia. The study will be conducted in accordance with ethical principles that comply with the Declaration of Helsinki and the International Council on Harmonisation's Harmonised Tripartite Guideline for Good Clinical Practice (GCP). All participants will provide written informed consent or parental assent prior to entering a study.

### WIC programme—management

The University of Warwick is the sponsor organisation. A programme manager (PM) is responsible for the day-to-day management of the research programme and supports the GHRG director for the overall delivery of the research programme. The PM, GHRG director and WP leads are responsible for the overall management of the

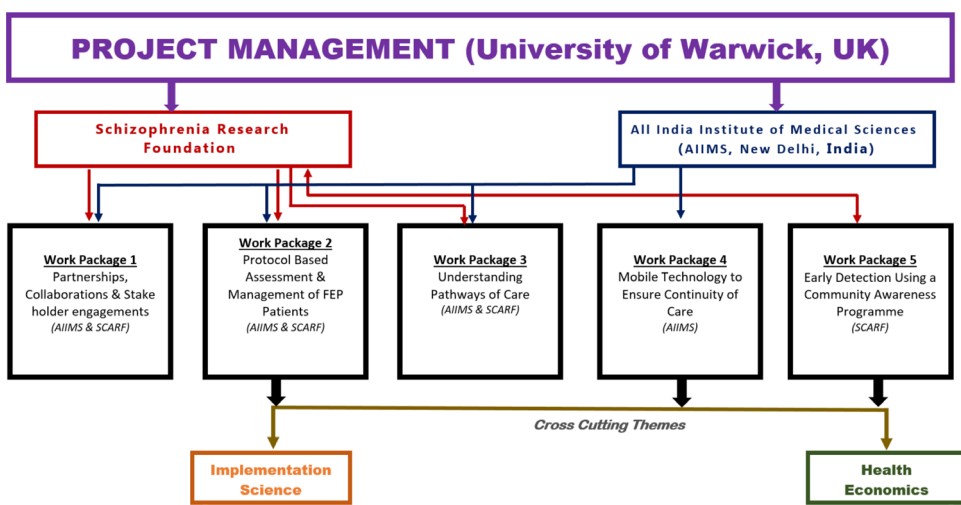

**Figure 1** Schematic overview of Warwick-India-Canada Global Health Research Group. AIIMS, All India Institute of Medical Sciences; FEP, first episode of psychosis; SCARF, Schizophrenia Research Foundation.

study and meet monthly by video conference chaired by the director. The programme steering committee (PSC), comprising PM, GHRG director, co-investigators and WP leads, provides overall strategic direction, ensures good governance, provides scientific input, guides capacity-building activities and provides financial probity when required. An independent external scientific advisory group (ESAG) comprising five international experts will monitor and assess the study conduct and progress. Both PSC and ESAG meet annually with the consortium to assess the scientific quality of research and to give feedback.

## Study sites

WIC brings together four institutions: University of Warwick, UK; McGill University, Canada; All India Institute of Medical Sciences (AIIMS), New Delhi, India; and Schizophrenia Research Foundation (SCARF), Chennai, India. SCARF, based in Chennai, is a non-profit voluntary organisation providing care to patients with psychosis and a WHO-collaborating centre for schizophrenia research. With a population of over 7 million, Chennai is the fourth largest metropolis in India. AIIMS, based in New Delhi with its metropolitan area population of 30 million, is a tertiary care hospital involved in providing undergraduate and postgraduate medical education and research, and hosts the National Drug Dependence Treatment Centre, a WHO-collaborating centre.

## WP1: partnerships, collaborations and stakeholder engagement

In WP1, we developed a network of external stakeholders in youth mental health including user and carer advocacy groups, clinicians, academicians and policy-makers, and it will be conducted at both study sites. A situational analysis and needs analysis were conducted to identify user and carer priorities, barriers to help-seeking and modifiable aspects of care pathways. 'Therapeutic ingredients' of early intervention that can be introduced in clinical care

in India will be identified. The results from this WP have been published elsewhere.[25 26]

## WP2: protocol-based assessment and management of FEP

Using the insights from WP1 and evidence-based interventions developed in the UK and Canada for EIS, in WP2 we will adapt protocols from the UK NEDEN study,[27] the largest FEP cohort in the world, to gather FEP cases at both SCARF and AIIMS, ensuring valid and reliable translation of assessment tools. A comphrehensive package of biopsychosocial care, tailored to the local context, will be developed and delivered to all new FEP cases and their short-term and medium-term outcomes measured. An FEP patient registry (comprising both affective and non-affective psychosis) will be maintained at our study sites. The study population, study assessment tools and study assessment schedules are described in Section A in the online supplemental material.

## WP3: understanding pathways to care

The objective of WP3 is to determine pathways to care in FEP at both sites, using a combination of qualitative and quantitative approaches. We will explore DUP and its components (help-seeking delay, treatment delay following presentation), cultural attribution of psychosis symptoms (eg, individual, supernatural), help-seeking behaviour and modifiable factors that contribute to DUP. We will exploit the established methodology for case ascertainment and care pathways from the NIHR ENRICH Programme.[28] We will examine the following questions:

1. Do the two cohorts (South India vs North India) differ significantly in culturally mediated illness attributions (ie, individual, natural, social, supernatural and no attribution) during FEP? The cohorts vary demographically and in terms of study setting (outpatient non-governmental organisation in South India vs large

general hospital with emergency, inpatient and outpatient facilities in North India).

2. Do the two cohorts differ in their pathways to care (ie, DUP, compulsory detention, psychiatrists, emergency services, general practitioner, faith-based organisations) during FEP?

3. Are the pathways to care during FEP influenced by culturally mediated illness attributions and other relevant sociodemographic (ie, gender, religious practice, marital status, living alone, education, income) and clinical confounders such as DUP?

### Study setting, population and study assessment tools

This study will be conducted at SCARF, Chennai, and AIIMS, New Delhi, using the data sets collected from FEP patients recruited as part of WP2. The study instruments that will be used in this study are (1) the *Emerging Psychosis Attribution Schedule*,[29] (2) Nottingham Onset Schedule[30] and (3) an amended version of the *Encounter form* by Gater and Sousa.[29]

### WP4: mobile technology to ensure continuity of care

Globally, India has the second largest number of mobile phone users—900 million users accounting for 88% of all telecommunication users.[31] Studies have emerged from high-income countries on mobile technologies in psychosis[32] but are lacking in LMICs.[33] Our preliminary survey shows that 88% of patients have mobile phones.[34] The objective of WP4 is to develop a home-based psychosocial care model for 'difficult-to-treat' patients with schizophrenia aimed at functioning and recovery, and an associated mobile-based application. In the absence of state provision in India, mobile-based clinical care has the potential to facilitate appropriate clinical care. In this WP, we will compare and evaluate the feasibility and effectiveness of a paper version of the model with the mobile app in improving patient outcomes and reducing carer burden. This study will be conducted at AIIMS, New Delhi. The study population, inclusion/exclusion criteria, methodology, assessment tools and study assessment schedules are described in Section B of the online supplemental material.

### WP5: early detection using a community awareness programme

The primary aim of WP5 is to develop and implement a school-based and college-based intervention for early detection and referral of individuals with emerging mental health problems. We wish to pilot a school-based and college-based programme of online screening for emerging mental health problems, assess the effectiveness of using community resources (teachers, peers and youth groups) to provide psychosocial intervention and determine the clinical effectiveness and cost-effectiveness of the screening programme. This study will be conducted by SCARF in Chennai, India. The identified catchment area for the study is Anna Nagar, Chennai, and the adjacent areas have 14 schools and 4 colleges. Informed consent from the management of the educational institutions will

be sought before conducting the study at their institution. The study assessment tools and study intervention are described in Section C of the online supplemental material.

### Two CTs

#### *Implementation science*

Despite being the first LMIC to introduce a National Mental Health Programme, there remains a significant implementation gap in the delivery of evidence-based interventions for mental healthcare in India.[35 36] Implementation science represents an approach focused on understanding the challenges, identifying the barriers and harnessing facilitators to help translate research-based evidence to 'real-world' clinic and community settings, developing leadership and changing professional practice to support delivery of evidence-based innovation.[37]

The primary objective of this study is to evaluate the implementation of the EIS in India. We will address the implementation barriers and facilitators in healthcare according to the Consolidated Framework for Implementation Research.[38] Our specific aims are (1) to identify barriers and facilitators to implementation, sustaining (beyond pilot) and scaling up of evidence-based early intervention to address psychosis and (2) to assess how the original intervention is adapted to fit context and to what extent it then diverges from its original evidence base. The study will be carried out at SCARF, Chennai.

#### *Health economics*

The primary objective of the health economics theme is to assess the cost of treatment for patients with FEP attending AIIMS from the payer and provider perspective. We will develop and pilot a questionnaire to determine direct and indirect costs experienced by patients with mental illness, and their households, and the coping strategies used in response to these costs. From these data, we will estimate mean, median and quartile costs at each time point. We will also conduct a bottom-up costing of the typical package of care offered to those participating in WP2 at AIIMS. This will be based on unit cost data collected during the study, along with interviews with clinicians taking part in the study on the care delivered and the time taken to deliver it.

#### *Capacity-building activities*

There is an urgent need to develop and deliver high-quality research on capacity-building activities in India, ranging from modules on research methodology to doctoral-level training and courses aimed at healthcare and leadership.

#### PhDs

WIC, along with TATA Trust, India, and Warwick Medical School, is supporting three PhD candidates from India at the University of Warwick by enabling them to carry out their primary research activities in their home countries. These three students, who were selected after a competitive process, commenced their doctoral studies

in May 2019. The PhD topics included projects that are nested within WIC and stand-alone projects that are closely aligned to the WIC research priorities. At least two University of Warwick-based supervisors and one partner country-based supervisor are allocated to each student for their PhDs. Topics approved for the first cohort of WIC PhD students include (1) health economics, (2) pathways of care and service engagements in early intervention services and (3) assessing factors associated with hospital readmissions and strategies to reduce the burden of readmissions in psychiatric hospitals in India.

### Research leadership training programme

At Warwick Business School, we already have an international distance-learning *Leadership Development Programme* for clinicians, with case studies from Africa and South East Asia. We will develop a contextualised distance-learning intervention about leadership and innovation for mental health professionals in India. This innovative training will be focused on developing another case from the WIC programme, specific to youth mental health in India. Integrating leadership and innovation training in clinical and service development domains will be beneficial for mental health professionals, particularly in formulating clinical priorities and engaging meaningfully with policy-makers in the country.

### Planned extensions

The WIC GHRG has secured funding from NIHR to conduct an additional programme of work, which builds on the existing programme of work, for a duration of up to 12 months. In this costed extension programme, we will (1) extend our current school-based screening for mental disorders (WP5) in two vulnerable populations: (a) refugee youth (Jammu & Kashmir) and (b) a high-risk tribal population in Mahrashtra; (2) expand our FEP registry (WP2) to a rural setting (Nagpur), for potential comparisons with other sites and with existing data sets and including additional research questions of interest (such as cardiometabolic parameters, lifestyle interventions); (3) extend health economic analysis to include rural populations and explore how financial burden and coping mechanisms differ between settings; and (4) develop a research-based leadership training programme.

### Analysis plan

For WP1, a prospective cross-sectional study in the form of a situational analysis and needs assessment with mental health professionals and external stakeholders will be conducted to capture information on current practices and available resources for the management of FEP. A semistructred pro forma will be developed to capture quantitative and qualitative data on facilities available for FEP management at each health centres, pharmacological and non-pharmacological management, local training needs, hospitalisation data and follow-up data of FEP cases. The data will be analysed using standard summary statistical methods, with indicators of variability,

central tendency, frequency, percentage, ratio or proportion, as appropriate

For the remaining WPs and CTs, we will use standard descriptive statistical methods to evaluate the data. Where appropriate, continuous variables will be assessed for normality, and non-normally distributed variables will be transformed or assessed using non-parametric tests as appropriate. To compare the data (eg, sociodemographic and clinical factors) between and within study sites, we will use independent-samples t-tests for continuous variables and $\chi^2$ for dichotomous variables (WP2 and WP3). To assess the change in several domains of symptom severity and functioning at prespecified time points, we will use a general linear modelling analysis, adjusting for various potential confounders (eg, WP2, WP4). We will use test-adjusted and test-unadjusted associations using statistical tests as appropriate (eg, WP3). For unadjusted analysis, we will use (1) analysis of variance for testing unadjusted associations between ethnic groups and culturally mediated illness attributions and service encounters (normally distributed continuous outcomes), (2) Kruskal-Wallis rank-sum tests (for non-normally distributed continuous outcomes) and (3) logistic regression for dichotomous variables. For adjusted association analysis, we will use multiple logistic regression analysis, controlling for various confounders of interest. To provide information regarding the magnitude of the change in scores in domains of interest (eg, WP5), *Cohen's d* effect sizes will be calculated with effect size values of ≥0.5, ≥0.8 and ≥1.0 corresponding to moderate, large and very large effect, respectively (eg, WP5). For CTs such as implementation science and health economics, we will use standard descriptive statistics to describe the data and framework analysis (implementation science), if deemed necessary.

Statistical significance will be inferred at $p < 0.05$. We will consider sensitivity to change as adequate, if the improvement is statistically significant ($p < 0.05$) and with at least a moderate effect size (Cohen's d ≥0.50). The primary analysis will be undertaken by excluding patients with missing data. However, we will conduct sensitivity analysis for missing data using the following imputation techniques, as appropriate: (1) last value carried forward, (2) baseline value carried forward, (3) single imputation with expectation–maximisation algorithm and (4) multiple imputation. Data will be analysed using appropriate statistical software (eg, SPSS, R, STATA).

### Assuring instrument reliability

Clinical instruments requiring ratings by research staff will continue to be tested for inter-rater reliability across the two sites, SCARF and AIIMS. A joint meeting of the research staff from both sites was held and inter-rater reliability exercises on the important measures were conducted using video-recorded interviews with 10 randomly selected patients with FEP. Data were successfully collected for the following measures: Scale for Assessment of Positive Symptoms, Scale for Assessment of Negative Symptoms, Young Mania Rating Scale,

Montgomery-Åsberg Depression Rating Scale, Brief Psychiatric Rating Scale, Global Assessment of Functioning, Social and Occupational Functioning Assessment Scale and Medication Adherence Rating Scale. A high degree of inter-rater reliability across sites (intraclass correlations from 0.74 to 0.92) was obtained on these measures. Our results were comparable with previous literature.[39 40]

## ETHICS AND DISSEMINATION
### Ethics
Written informed consent will be obtained from all participants or carers as appropriate, before their enrolment into the study. In the school-based and college-based study (WP5), all participants will receive information on where to seek help for mental health problems. In addition, we have developed a clinical care pathway for those who are identified as being at 'high risk' for severe mental illness or having undetected/untreated severe mental illness or with suicidal ideations that will ensure safety, protection of confidentiality and privacy. The study principal investigator and team will preserve the confidentiality and anonymity of participants taking part in the study and fulfil transparency requirements under the General Data Protection Regulation for health and care research. The study may be subject to inspection and audit by the University of Warwick under their remit as sponsor and other regulatory bodies to make sure of adherence to GCP and the UK Policy Framework for Health and Social Care Research.

This study has been approved by the University of Warwick's Biomedical and Scientific Research Ethics Committee (reference: REGO-2018-2208), Coventry, UK and research ethics committees of all participating organisations. The ethical conduct of the study is monitored throughout by the University of Warwick and each participating organisation.

### Dissemination strategies
We will strategically exploit our strong network with various LMIC stakeholders involved in mental health services such as clinicians, academics, service providers, policy-makers, government and non-governmental agencies, community groups, patient and carer advocacy groups, media agencies and the creative arts industry to support WIC dissemination activities. We will use a variety of methods to ensure effective dissemination of the groups' work, including (1) development of an up-to-date project website (to target different audiences/stakeholders); (2) presentations at national and international psychiatric and mental health scientific conferences, and peer review publications; and (3) publication through visual media including film and theatre. Dissemination activities through cinema and theatre have a particular resonance for many LMIC populations, and we believe that we will reach a large number of people through these activities. We will work closely with our LMIC partners to identify the dissemination approaches that might work best in this setting.

### Patient and public involvement
We have strong and ongoing engagement with various internal and external stakeholders involved in mental healthcare. The WP1 of WIC uses focus groups as a method for situational analysis and needs analysis and will include patients and carer groups, and voluntary/charity/third-sector organisations, ensuring representation of hard-to-reach groups. Patient and/or public groups were not involved in proposing the research objectives nor were involved in the overall study design or the implementation of this research programme. However, stakeholders involved in mental health will be engaged by the research team at respective sites as local dissemination plans are being devised. All partners also have an extensive record of patient and public involvement (PPI) in research. We have a robust programme for monitoring and governance—research, ethics, and financial, and our programme will adhere to the highest levels of ethical practice and probity including PPI involvement.

## CONCLUSION
Although major improvements have been achieved in the treatment of patients with psychotic disorders in high-income countries, including the wide-scale implementation of EIS in psychosis, implementing such services remains a major challenge in LMICs. The sizeable differences in patient profiles, sociodemographics, pathways of care, barriers to care, comorbidities, scarcity of resources and delayed presentation mean, at the very least, Western models of care cannot be simply translocated to LMICs. The WIC programme will tailor evidence-informed interventions to the Indian sociocultural context to address the increasing burden of psychotic disorders in India. By using a bespoke interventional approach that is regionally relevant and by exploiting the untapped potential of digital and mobile technologies in India, WIC will develop and test interventions that can be integrated into existing clinical care and generate data (including health economic data) that will inform priorities for healthcare providers and policy-makers.

**Author affiliations**
[1]Division of Mental Health & Wellbeing, Warwick Medical School, University of Warwick, Coventry, UK
[2]Coventry and Warwickshire Partnership Trust, Coventry, UK
[3]Prevention and Early Intervention Program for Psychosis (PEPP-Montréal), Douglas Mental Health University Institute, Verdun, Quebec, Canada
[4]Department of Psychiatry, McGill University, Montreal, Québec, Canada
[5]WMG and Warwick Medical School, University of Warwick, Coventry, UK
[6]Warwick Business School, University of Warwick, Coventry, UK
[7]Warwick Clinical Trials Unit, Warwick Medical School, University of Warwick, Coventry, UK
[8]Department of Psychiatry, All India Institute of Medical Sciences, New Delhi, Delhi, India
[9]Schizophrenia Research Foundation, Chennai, Tamil Nadu, India
[10]Institute of Applied Health Research, University of Birmingham, Birmingham, UK

**Acknowledgements** We thank the WIC Independent External Scientific Advisory Group (ESAG) for providing strategic insight and international perspectives for WIC and also for their valuable inputs to the critical decisions affecting the scientific orientation of the WIC research programme.

**Collaborators** Dr Pushpendra Singh (Indraprastha Institute of Information Technology, New Delhi), Ms Anupriya Tuli (Indraprastha Institute of Information Technology, New Delhi), Mr Sachin Chaudhary (Trust Circle, Chennai), Ms Tasneem Raja (TATA Trusts, Nagpur), Mr Ramesh Hangloo (Pir Panchal, Jammu & Kashmir), Dr Arti Bakshi (University of Jammu, Jammu & Kashmir), Dr Manu Arora (Government Medical College, Jammu & Kashmir), Ms Neeru Khera (Creative Gypsy, New Delhi). External Scientific Advisory Group (ESAG): Prof Aileen Clarke (University of Warwick), Prof Tom Burns (University of Oxford), Prof Sanjeev Jain (NIMHANS, Bangalore), Prof Mohan Isaac (University of Western Australia, Australia) and Prof Craig Morgan (King's College London).

**Contributors** SPS is the Director of this global health research group. SPS conceived the original study design and obtained funding. MM is the programme manager and prepared the first draft and subsequent revisions of the manuscript, and is the joint first author with SPS. TR,PR, MS, JM, GC and RKC contributed writing to particular sections relevant to their work package of the Group. SI,CM,JS,MB, RL and VF provided critical comments during the proposal-writing stage and WIC project meetings, and critically reviewed and revised the manuscript. All authors read and approved the final manuscript.

**Funding** This study/project is funded by the National Institute for Health Research (NIHR) (NIHR Group on Psychosis Outcomes: the Warwick-India-Canada (WIC) Network (Award number: 16/137/ 107)). SPS and MB are supported by the NIHR Applied Research Centre (ARC) West Midlands. SNI is supported by a salary award from the Canadian Institutes of Health Research. JS is supported by a clinician-scientist salary award from the Fonds de Recherche du Québec—Santé.

**Disclaimer** The views expressed are those of the author(s) and not necessarily those of the NIHR or the Department of Health and Social Care.

**Competing interests** None declared.

**Patient consent for publication** Not required.

**Provenance and peer review** Not commissioned; externally peer reviewed.

**ORCID iDs**
Swaran P Singh http://orcid.org/0000-0003-3454-2089
Mohapradeep Mohan http://orcid.org/0000-0003-1665-2081
Jason Madan http://orcid.org/0000-0003-4316-1480
Padmavati Ramachandran http://orcid.org/0000-0002-0380-8617

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
