## [Reviewer comments · BMJ Open]

ARTICLE DETAILS

TITLE (PROVISIONAL)	The Warwick-India-Canada (WIC) Global Mental Health Group: Rationale, Design and Protocol.
AUTHORS	Singh, Swaran; Mohan, Mohapradeep; Iyer, Srividya; Meyer, Caroline; Currie, Graeme; Shah, Jai; Madan, Jason; Birchwood, Max; Sood, Mamta; Padmavati, Ramachandran; Chadda, Rakesh; Lilford, RJ; Furtado, Vivek; Thara, Rangaswamy

VERSION 1 – REVIEW

REVIEWER	Cleverley, Kristin University of Toronto, Nursing
REVIEW RETURNED	22-Dec-2020

GENERAL COMMENTS	Thank you for having me review this important project. The studies within this project will have important and meaningful differences in the lives of an extremely vulnerable population. After careful review of this protocol I have no concerns or major edits. It is a thoughtful and well written protocol. Two minor queries that I will leave to authors to decide whether to update. 1. In WP2, how will exclusion criteria be determined and/or assessed? I.e. learning disability.2. In WP2, some indication of time to complete the study assessment package would be beneficial, either from pilot testing or previous studies.
--

REVIEWER	Vanderkruik, Rachel University of Colorado Boulder, Psychology and Neuroscience
REVIEW RETURNED	31-Jan-2021

GENERAL COMMENTS	The work of the WIC global mental health group is very important and the rationale, design and protocol is well described in this manuscript. I think it would be a valuable and important contribution to publish this protocol. A few comments: - Line 36/page 4, is "nonetheless" the intended word to start this sentence? Seems like "Furthermore" or a similar word would be more appropriate- I was interested to learn more of how the WIC network was established - what brought these countries together to address this important work? Maybe a line or so would be of interest to the reader.- The grammar could be improved in a few places throughout the manuscript (e.g., the structure of line 10/page 5 could flow better).- For the section of "The Indian Context" - I do wonder how this work compares to/builds from work in other countries with limited
---

	mental health resources and high stigma of mental health disorders.  - Regarding the objectives of the WIC to develop/culturally tailor interventions for individuals with first-episode psychosis, I'm wondering if more can be described regarding what interventions you will build/adapt from? - Regarding work Package 1 - how do you plan to establish such engagement with stakeholders in the context of stigma? Any challenges you foresee with attempts to engage youth/community stakeholders in this work? - The supplemental material contains a lot of information regarding program evaluation/measures - what about the content covered within the interventions? - Regarding work package 3 and understanding pathways to care - I wonder about any mention/consideration of engaging peers to facilitate care, given some of the work done with task-sharing/shifting to peers or lay professionals in India. - Regarding the work package 4 description on page 8 - I was wondering about any description on the feasibility of a mobile app with this target population/setting. This is addressed later in supplemental material - I think it could be beneficial to add a line to this extent earlier on page 8. - Regarding one of the 2 cross-cutting themes (Implementation Science) - I think this section could be strengthened by some reference of an implementation science framework that will be used for this work. - Regarding Analysis plan - will any qualitative data be used in terms of evaluation/stakeholder engagement and intervention development? If so, there could be a brief description of the qualitative data analysis approach - Regarding ensuring instrument reliability - is it sufficient to establish IRR once with just 10 patients? Should anything be done to assess IRR over time with this study? - Regarding patient and public involvement (page 13) - I'm not quite clear about this approach; is this falling under a community based participatory research (CPBR) approach? If so, as mentioned above, I'm wondering about any challenges anticipated with this work and how the partnership will address these, particularly given the stigma of severe mental health issues - Page 14 - I could not quite read the last full paragraph - text was jumbled. - Figure 1 is repeated? - Figure 2 could be clarified - not clear whether difficulty level to treat patients w/ schizophrenia is increasing or decreasing as go up triangle... - As mentioned above, in the "intervention" section Under table 4, more description on the content of the intervention would be of interest.
--	--

VERSION 1 – AUTHOR RESPONSE

Reviewer # 1: "Thank you for having me review this important project. The studies within this project will have important and meaningful differences in the lives of an extremely vulnerable population. After careful review of this protocol I have no concerns or major edits. It is a thoughtful and well written protocol"

We are very grateful for the positive comments of the reviewer.

Two minor queries that I will leave to authors to decide whether to update.

1. In WP2, how will exclusion criteria be determined and/or assessed? I.e. learning disability.

The exclusion criteria such as learning disability (LD) will be ascertained by: (i) previous diagnosis of LD ; or (ii) by trained psychologist / psychiatrist through detailed history from the patients and their caregivers, corroborated by observations from academic performance assessed using test scores from various subjects (such as language and mathematics).

2. In WP2, some indication of time to complete the study assessment package would be beneficial, either from pilot testing or previous studies.

The estimated time to complete the study assessments will be around 60 min-90 min. We plan to conduct each assessment in two or more sessions depending on the participant's time, availability and comfort.

We have made the following changes to the manuscript to address these recommendations:

For Query 1: We have now revised Section A, WP-2 – Study Population (page 1), of supplementary material.

For Query 2: We have now revised Section A, WP-2 – Study Assessment Tools (page 2), of supplementary material.

Reviewer # 2: "The work of the WIC global mental health group is very important and the rationale, design and protocol is well described in this manuscript. I think it would be a valuable and important contribution to publish this protocol. A few comments":

We thank the reviewer for this endorsement.

1. Line 36/page 4, is "nonetheless" the intended word to start this sentence? Seems like "Furthermore" or a similar word would be more appropriate

We apologise for this error. We have now re-written this sentence (page 4, line 25) and it now read as " Furthermore, DUP is significantly higher in LMICs ¹⁶, and therefore EIS provision is potentially of great value".

2. I was interested to learn more of how the WIC network was established - what brought these countries together to address this important work? Maybe a line or so would be of interest to the reader.

We agree with the reviewer for this valuable suggestion. We have now added the following sentence to the Aim and Strategic Goals section of manuscript (page 5, line 5):

“The WIC network brings together knowledge and expertise of four internationally recognised institutions that share research interests, expertise in effective interventions in psychotic disorders, strategic vision to reduce the burden of psychotic disorders in resource-poor settings, and who are keen to expand their research knowledge to implement evidence-based clinical changes in LMIC settings”.

3. *The grammar could be improved in a few places throughout the manuscript (e.g., the structure of line 10/page 5 could flow better).*

We apologise for grammatic errors. We have now revised the manuscript and have made appropriate changes. We hope the manuscript reads well now.

4. *For the section of "The Indian Context" - I do wonder how this work compares to/builds from work in other countries with limited mental health resources and high stigma of mental health disorders.*

We thank the reviewer for pointing out that limited resources and high stigma of mental illness are not exclusive to the Indian context and we should build upon relevant research in other settings. We certainly want to build upon previous work on building capacity in resource-poor settings and stigma reduction. In this project we wish to demonstrate that the principles of successful early intervention programmes are, in principle, implementable in poorly resources countries.

When describing the findings of this work, we will be mindful of the reviewer’s

comments and interpret these in light of previous work on maximising limited resources, for instance the task shifting paradigm

(Joshi et al 2014 <https://journals.plos.org/plosone/article?id=10.1371/journal.pone.0103754>)

5. *Regarding the objectives of the WIC to develop/culturally tailor interventions for individuals with first-episode psychosis, I'm wondering if more can be described regarding what interventions you will build/adapt from?*

We have now amended the “Objectives” section (page 5, key objective. a) and will read as follows: “develop a culturally tailor-made FEP management protocol, based on evidence synthesis of current clinical practice at partners sites and by modifying protocol from the NEDEN study, to make sustained health improvements for poor served individuals with FEP.”

6. *Regarding work Package 1 - how do you plan to establish such engagement with stakeholders in the context of stigma? Any challenges you foresee with attempts to engage youth/community stakeholders in this work?*

We completely agree with Reviewer 2 that establishing a meaningful engagement with various stakeholders involved in mental healthcare in LMIC communities is a huge challenge, especially in the context of stigma. Our extensive local, regional, national and international networks with our collaborators have helped us to develop a list of key stakeholders in India, including patient and carer advocacy groups, voluntary, charity and third sector organisations, human right organisations, nongovernmental organisations and community services active in mental health care and reform. Additionally, two members of the WIC external scientific advisory group (ESAG) have substantial track record and influence in mental health research programmes in India. For instance, Prof Mohan Issac led the Indian National Mental health programme and Prof Sanjeev Jain has been a national leader in community provision for psychotic disorders. All these networks has hugely helped us in

overcoming the anticipated challenges, as well as in providing a nuanced understanding of the strengths of Indian communities, assets and untapped potential. We are delighted to inform you that we have now completed this work package and have published our findings in the journal of Early Intervention in Psychiatry.

Publications from WP-1:

- (1) V. R. Dhandapani, S. Chandrasekaran, S. Singh, M. Sood, et.al : Community stakeholders' perspectives on youth mental health in India: Problems, challenges and recommendations. Early Interv Psychiatry (2020) doi:10.1111/eip.12984.
- (2) V. R. Dhandapani, P. Ramachandran, G. Mohan, S. Chandrasekaran, et.al : Situational analysis of prevailing practices in the management of first-episode psychosis in Chennai, India. Early Interv Psychiatry (2020) doi:10.1111/eip.12979

7. *The supplemental material contains a lot of information regarding program evaluation/measures - what about the content covered within the interventions?*

We apologise for not providing sufficient information on the content covered within the interventions.

We have now updated the supplementary material to address reviewer's comment.

8. *Regarding work package 3 and understanding pathways to care - I wonder about any mention/consideration of engaging peers to facilitate care, given some of the work done with task-sharing/shifting to peers or lay professionals in India.*

Our aim in this programme was to determine whether early intervention can be delivered by **existing mental health care providers** in India, in two different settings - a tertiary hospital with a psychiatry department in North India and an NGO in South India. The reviewer is absolutely correct that in resource poor settings, alternative ways to provide mental health care are through task shifting (lay professional, community workers, peer supporters etc). We plan to explore that question in our future projects. In this paper we are describing the addition of early intervention components to services offered by existing providers.

9. *Regarding the work package 4 description on page 8 - I was wondering about any description on the feasibility of a mobile app with this target population/setting. This is addressed later in supplemental material - I think it could be beneficial to add a line to this extent earlier on page*

8.

We have now rewritten the "Work Package 4:- Mobile technology to ensure continuity of care" section (page 9) of the manuscript to address reviewer's comment.

10. *Regarding one of the 2 cross-cutting themes (Implementation Science) - I think this section could be strengthened by some reference of an implementation science framework that will be used for this work.*

We thank the reviewer for this suggestion. This cross-cutting theme will address the implementation barriers and facilitators in healthcare according to the

Consolidated Framework for Implementation Research (CFIR). We have updated this section (page 10, Implementation Science), to address reviewer's comment.

7. Regarding Analysis plan - will any qualitative data be used in terms of evaluation/stakeholder engagement and intervention development? If so, there could be a brief description of the qualitative data analysis approach

For WP-1 (evaluation/stakeholder engagement), a prospective cross-sectional study in the form of a situational analysis and needs assessment with mental health professionals and external stakeholders will be conducted to capture information on current practices and available resources for the management of FEP. The data collected will be both quantitative and qualitative. We have amended “ analysis plan” section (page 12) for WP-1, to address the reviewer's comment.

8. Regarding ensuring instrument reliability - is it sufficient to establish IRR once with just 10 patients? Should anything be done to assess IRR over time with this study?

Thank you for this comment. Our decision to use 10 randomly selected cases for ascertaining the IRR between the researchers within and across site was based on previous published literature (below) from our study sites. Importantly, our results were comparable with previous literature.

- Iyer SN, Mangala R, Thara R, et al. Preliminary findings from a study of first-episode psychosis in Montreal, Canada and Chennai, India: comparison of outcomes. *Schizophr Res* 2010;121(13):227-33. doi: 10.1016/j.schres.2010.05.032
- Malla A, Iyer SN, Rangaswamy T, et al. Comparison of clinical outcomes following 2 years of treatment of first-episode psychosis in urban early intervention services in Canada and India. *Br J Psychiatry* 2020:1-7. doi: 10.1192/bjp.2020.126 [published Online First: 2020/07/06]

Regarding IRR overtime: All our research staffs at both sites were rigorously trained as part of this study and have extensive experience in undertaking similar researches. At both sites, data were collected only for one year assessments. While we recognise that IRR can change over time, taking into account of our extensive experience in undertaking similar research, we do not anticipate any meaningful changes to occur within an year. However, we intend to undertake IRR retest for our planned extension activities (as described in page 12 of manuscript), to minimise the possibility that recall of the first assessments would influence the second assessments.

11. Regarding patient and public involvement (page 13) - I'm not quite clear about this approach; is this falling under a community based participatory research (CPBR) approach? If so, as mentioned above, I'm wondering about any challenges anticipated with this work and how the partnership will address these, particularly given the stigma of severe mental health issues

We apologise for the confusion here. We would like to clarify that our research do not fall completely under community-based participatory research (CBPR) approach. This is primarily because the researchers and the stakeholders do not engage as equal partners in all steps of our multi-component research programme. However aspects of CBPR such as stakeholder engagement as part of our WP1/WP-4, have complemented or enhanced our PPI plan. This approach was needed for these WPs because they had the benefit of making sure a direct involvement of patients and service users rather than listening to professional voices as a proxy for patient experience, especially while

developing a culturally tailor-made protocol for FEP management and developing a bespoke module for home based psychosocial care model for continuity of care.

To avoid any confusion to the readers, we have revised the 'Patient and Public Involvement' section of the manuscript.

12. Page 14 - I could not quite read the last full paragraph - text was jumbled.

We think this was a technical glitch in the submission portal. We noticed that the texts in the same section (Page 16 & 38 of manuscript, clean version and marked copy respectively) still appears jumbled. We tried different ways to resolve this (creating new word document etc), however was in vain. We will highlight this to the editor in our resubmission process and seek assistance from the IT department of the journal to resolve this.

The jumbled text read as follows:

"Disclaimer: This research was commissioned by the National Institute of Health Research using Official Development Assistance (ODA) funding. The views expressed are those of the author(s) and not necessarily those of the NIHR or the Department of Health and Social Care.

Funding: United Kingdom's National Institute of Health Research (NIHR) using Official Development Assistance (ODA) funding (Award number: 16/137/107)".

15. Figure 1 is repeated?

We apologise for this error. We included figure 1 in manuscript as well as a separate file during our submission process. Hence, the figure was repeated. We will ensure the figures are not repeated in our resubmission.

15. Figure 2 could be clarified - not clear whether difficulty level to treat patients w/ schizophrenia is increasing or decreasing as go up triangle

We apologise for the confusion with figure 2 (renamed now as supplementary figure 1). The total number of patients is same in all three arms of the study. To avoid confusion, we have now changed this figure to simpler one.

16. As mentioned above, in the "intervention" section Under table 4, more description on the content of the intervention would be of interest.

We apologise for not providing sufficient information on the content covered within the interventions. We have now updated the supplementary material to address reviewer's comment.